# Microbe-Derived Antioxidants Reduce Lipopolysaccharide-Induced Inflammatory Responses by Activating the Nrf2 Pathway to Inhibit the ROS/NLRP3/IL-1β Signaling Pathway

**DOI:** 10.3390/ijms232012477

**Published:** 2022-10-18

**Authors:** Cheng Shen, Zhen Luo, Sheng Ma, Chengbing Yu, Qingying Gao, Meijuan Zhang, Hongcai Zhang, Jing Zhang, Weina Xu, Jianbo Yao, Jianxiong Xu

**Affiliations:** 1Shanghai Key Laboratory of Veterinary Biotechnology, School of Agriculture and Biology, Shanghai Jiao Tong University, Shanghai 200241, China; 2Division of Animal and Nutritional Sciences, West Virginia University, Morgantown, WV 26506, USA

**Keywords:** microbial-derived antioxidants, oxidative stress, inflammatory response, Nrf2, ROS/NLRP3/IL-1β

## Abstract

Inflammation plays an important role in the innate immune response, yet overproduction of inflammation can lead to a variety of chronic diseases associated with the innate immune system; therefore, modulation of the excessive inflammatory response has been considered a major strategy in the treatment of inflammatory diseases. Activation of the ROS/NLRP3/IL-1β signaling axis has been suggested to be a key initiating phase of inflammation. Our previous study found that microbe-derived antioxidants (MA) are shown to have excellent antioxidant and anti-inflammatory properties; however, the mechanism of action of MA remains unclear. The current study aims to investigate whether MA could protect cells from LPS-induced oxidative stress and inflammatory responses by modulating the Nrf2-ROS-NLRP3-IL-1β signaling pathway. In this study, we find that MA treatment significantly alleviates LPS-induced oxidative stress and inflammatory responses in RAW264.7 cells. MA significantly reduce the accumulation of ROS in RAW264.7 cells, down-regulate the levels of pro-inflammatory factors (TNF-α and IL-6), inhibit NLRP3, ASC, caspase-1 mRNA, and protein levels, and reduce the mRNA, protein levels, and content of inflammatory factors (IL-1β and IL-18). The protective effect of MA is significantly reduced after the siRNA knockdown of the NLRP3 gene, presumably related to the ability of MA to inhibit the ROS-NLRP3-IL-1β signaling pathway. MA is able to reduce the accumulation of ROS and alleviate oxidative stress by increasing the content of antioxidant enzymes, such as SOD, GSH-Px, and CAT. The protective effect of MA may be due to its ability of MA to induce Nrf2 to enter the nucleus and initiate the expression of antioxidant enzymes. The antioxidant properties of MA are further enhanced in the presence of the Nrf2 activator SFN. After the siRNA knockdown of the Nrf2 gene, the antioxidant and anti-inflammatory properties of MA are significantly affected. These findings suggest that MA may inhibit the LPS-stimulated ROS/NLRP3/IL-1β signaling axis by activating Nrf2-antioxidant signaling in RAW264.7 cells. As a result of this study, MA has been found to alleviate inflammatory responses and holds promise as a therapeutic agent for inflammation-related diseases.

## 1. Introduction

As a defense mechanism, inflammation occurs when an organism is threatened with many harmful stimuli, including damaged cells or invading pathogens and endotoxins, and plays a significant role in the immune response to pathogenic infections [1]. There has been evidence suggesting chronic inflammation is associated with chronic diseases of the innate immune system, including rheumatoid arthritis, cardiovascular disease (CVD), and inflammatory bowel disease (IBD) [2,3,4]. Therefore, modulating excessive inflammatory responses has been an effective strategy for treating inflammatory diseases. Lipopolysaccharide (LPS) is an exogenous endotoxin found in Gram-negative bacteria’s outer cell membrane, which causes inflammation. It has been extensively used to stimulate inflammation in previous studies [5]. Among the significant immune cells involved in the inflammatory response, macrophages play an important role in initiating, maintaining, and resolving inflammation [1]. When stimulated by LPS, macrophages become polarized and activate various signaling pathways, including mitogen-activated protein kinase (MAPK) and nuclear factor-κB (NF-κB), resulting in the production of inflammatory mediators and pro-inflammatory cytokines such as tumor necrosis factor-α (TNF-α) and interleukin-1 beta (IL-1β) [6,7]. As a result, LPS-stimulated RAW264.7 cells are commonly used to evaluate anti-inflammatory properties.

Oxidative stress is a condition in which an imbalance exists between the production of free radicals and reactive metabolites (described as oxidants or reactive oxygen species (ROS)) and their elimination by protective mechanisms (also known as antioxidants). It is widely accepted that inflammation is associated with oxidative stress and that elevated levels of intracellular ROS are considered to be the most potent mediators of inflammation [8]. LPS-activated macrophages lead to excessive production of ROS [5]. ROS overproduction also functions as a signal to activate the nucleotide-binding domain (NOD)-like receptor protein 3 (NLRP3) inflammasome [9]. During activation of the NLRP3 inflammasome, the NLRP3 protein polymerizes and binds to the ASC adapter, which activates the cysteine protease caspase-1. When activated, caspase-1 creates precursors of the pro-inflammatory cytokines IL-18 and IL-1β, which are then matured and secreted to participate in the resulting inflammatory response [9,10]. Studies have shown that reducing ROS inhibits the activation of NLRP3, indicating that oxidative stress plays a significant role in inflammation [5,11]. Nuclear factor (erythroid-derived-2)-like 2 (Nrf2) translocates to the nucleus and regulates the expression of antioxidant and anti-inflammatory factors in cells [11,12]. When Nrf2 is activated, it can dissociate from the Kelch-like ECH-associated protection 1 (Keap1) and translocate to the nucleus, where it binds to antioxidant response elements (AREs). These AREs regulate the expression of antioxidant genes, such as NAD(P)H quinone oxidor-eductase-1 (NQO-1), and heme oxygenase 1 (HO-1) [13]. There is growing evidence that activation of Nrf2 blocks ROS overproduction and thus inhibits NLRP3 inflammasome activation. Inflammatory conditions can be effectively treated by targeting Nrf2 signaling [5,6,14].

Currently available anti-inflammatory drugs are steroids and non-steroids, which are associated with severe cardiovascular and gastric side effects, limiting their clinical use [15]. Naturally fermented products for the treatment of chronic inflammatory conditions are becoming a hot research topic. Sea buckthorn and prickly pear are two highly valued plants known for their unique functional and nutritional components, particularly their outstanding antioxidant capacities and health benefits [16,17]. Microbe-derived antioxidants (MA) are a fermented mixture of sea buckthorn and prickly pear with probiotics. It contains numerous bioactive compounds, including isoflavones, glutathione, and selenium. According to our previous studies, MA has good free radical scavenging ability and improves weaning-induced stress in piglets [18,19,20]. It also significantly reduced IL-1β and IL-18 levels in serum and NLRP3, IL-1β, and IL-18 gene expression in the liver of female rats induced by high-fat diets [21]. However, the molecular mechanisms underlying the antioxidant and anti-inflammatory effects of MA are not fully understood. In the present study, we used mouse macrophages (RAW264.7 cells) as a cell model and constructed RAW264.7 cells with knockdown of Nrf2 and NLRP3 to investigate the molecular mechanisms of the effects of MA on oxidative stress and inflammatory responses.

## 2. Results

### 2.1. MA Exhibits Anti-Inflammatory Effects in LPS-Induced RAW264.7 Cells

Cell viability was reduced to 49.56% when RAW264.7 cells were treated with 1 mg/L of LPS compared to the control, while treatment with 100 mg/L of MA significantly protected RAW264.7 from LPS-induced cytotoxicity (Figure 1a). There was no significant difference in the enhancement of cell viability between 100, 200, and 500 mg/L MA, therefore 100 mg/L MA was used for subsequent experiments. The mRNA levels of TNF-α, IL-1β, IL-6, and IL-18 were significantly increased under LPS induction, and MA significantly inhibited this increase (Figure 1b). We also observed increased production of IL-1β and IL-18 protein levels under LPS induction, whereas IL-18 and IL-1β protein levels were significantly reduced when cells were simultaneously treated with MA (Figure 1c). A significant increase in the concentration of the inflammatory factors TNF-α, IL-1β, IL-6, and IL-18 was observed by the induction of LPS, and MA significantly inhibited this increase (Figure 1d).

### 2.2. MA Attenuates LPS-Induced Inflammatory Effects by Inhibiting NLRP3

NLRP3, ASC, and Caspase-1 gene expression levels and protein expression levels were significantly increased under LPS induction, and MA significantly inhibited this increase (Figure 2a,b). To further validate the role of the NLRP3 inflammasome in the LPS-induced inflammation, RNA silencing of NLRP3 experiments was performed. Transfection with NLRP3 siRNA resulted in a significant decrease in approximately 79% of NLRP3 mRNA expression and approximately 60% of NLRP3 protein expression (Figure 2c,d). NLRP3 siRNA transfection significantly reduced the LPS-stimulated increase in IL-18 and IL-1β mRNA and protein expression levels, and this reduction was further markedly enhanced by synergistic treatment with MA (Figure 2e,f). Under LPS stimulation, IL-1β decreased by 50.3% after NLRP3 siRNA transfection and by 61.5% when MA was added (Figure 2g). NLRP3 siRNA transfection significantly reduced the LPS-stimulated increase in NLRP3, ASC, and Caspase-1 mRNA levels and protein expression, and co-treatment with MA further significantly enhanced this reduction (Figure 2h,i). Overall, the results suggest that MA can attenuate LPS-induced inflammatory effects by inhibiting NLRP3 activation to suppress the NLRP3/IL-1β signaling pathway.

### 2.3. MA Mitigates LPS-Induced Oxidative Stress in RAW264.7 Cells

LPS-induced oxidative stress, specifically ROS production, is thought to be a key trigger for initiating NLRP3 inflammasome activation. Cells exposed to LPS for 24 h were observed to exhibit a more than 3-fold increase in intracellular ROS levels, while treatment with 100 mg/L MA and 10 μM NAC were both able to significantly inhibit the elevated ROS (Figure 3a). When treated with LPS, the activities of SOD, GSH-Px, and CAT decreased rapidly and the MDA levels increased rapidly, indicating a rapid decrease in the antioxidant capacity of the cells in vivo, a trend that was inhibited by both MA and NAC. The ROS scavenging effect of MA at 100 mg/L was comparable to that of NAC (10 μM) treatment, which is a recognized antioxidant (Figure 3b).

### 2.4. MA Inhibits LPS-Induced Cellular Inflammation through Activation of the Nrf2/NLRP3/IL-1β Signaling Pathway

Further significant reductions in NLRP3 and IL-1β mRNA levels and protein expression were observed in RAW264.7 cells following MA and NAC synergistic SFN(Nrf2 activator) treatment compared to the MA and NAC treatment groups alone (Figure 4a,b). Changes in IL-1β levels showed a similar trend (Figure 4c). Nrf2 is thought to prevent oxidative damage caused by inflammation or injury by regulating the expression of antioxidant genes. MA treatment significantly increased the levels of Nrf2 protein in the nucleus (Figure 4d), resulting in a significant increase in Nrf2, HO-1, and NQO1 mRNA levels and protein expression in RAW264.7 cells (Figure 4e,f), which was reinforced in the presence of the Nrf2 activator SFN. The transfer of activated Nrf2 from the cytoplasmic lysate to the nucleus induced transcription of many Nrf2-regulated antioxidant enzymes. Consistently, MA treatment significantly increased the activities of SOD, GSH-Px, and CAT, decreased MDA levels, reduced ROS content, and improved oxidative stress status. Treatment results in the NAC group were similar to those in the MA group (Figure 4g,h).

To further validate the role of Nrf2 in the LPS-induced NLRP3/IL-1β signaling pathway, Nrf2 silencing experiments were conducted. Approximately 73% of Nrf2 mRNA expression and approximately 72% of Nrf2 protein expression were significantly decreased in cells after transfection with Nrf2 siRNA (Figure 5a,b). In contrast to the results of treatment with the Nrf2 activator SFN, the inhibitory effect of MA on inflammatory responses via the Nrf2/NrRP3/IL-1β signaling pathway was significantly hindered by Nrf2 silencing (Figure 5c–g). When Nrf2 was silenced, the tendency of MA to promote elevated cellular antioxidant enzyme activity and reduce IL-1β secretion was significantly suppressed, ROS levels were significantly increased, and cells showed severe oxidative stress and inflammatory responses (Figure 5e,h,i). These findings suggest that the Nrf2 gene plays a regulatory role as a key node in the Nrf2/NLRP3/IL-1β signaling pathway and that MA can modulate the inflammatory response through the Nrf2/NLRP3/IL-1β signaling pathway.

## 3. Discussion

Pathophysiologically speaking, inflammation protects the host against an extensive range of pathogens. In response to stimuli, pro-inflammatory cytokines produced by macrophages have been proposed as significant factors in the development and progression of many inflammatory diseases [22,23]. Thus, blocking the release of inflammatory mediators might be a promising treatment for inflammatory diseases. Our results showed that RAW264.7 macrophages produced large amounts of ROS after LPS stimulation, leading to activation of the NLRP3 inflammatory factor signaling pathway to create the IL-1β inflammatory factor. MA treatment significantly reduced ROS levels and inhibited NLRP3 inflammatory factor activation, which in turn reduced IL-1β inflammatory factor. Further studies showed that this protective effect of MA was associated with the activation of NrF2 antioxidant signaling. MA induced translocation of NrF2 to the nucleus, thereby enhancing the activity of several antioxidant enzymes. Here, we demonstrate that MA can protect cells from LPS-induced cellular inflammation by inhibiting the ROS/NLRP3/IL-1β signaling axis via the Nrf2-antioxidant signaling pathway. The current study shows that MA inhibits the initiation phase of LPS-induced cellular inflammation, highlighting its potential ability to prevent and mitigate inflammation at an early stage.

It is well known that the NLRP3 inflammasome is a cytoplasmic multiprotein complex that plays an essential role in the development and progression of many inflammatory disorders [24]. LPS can be recognized by Toll-like receptors (TLRs), resulting in the motivation of the NF-κB signaling pathway. Once activated, NF-κB regulates the expression of pro-inflammatory cytokines, such as IL-1β. Once NLRP3 is activated, it recruits the bridging protein ASC, which promotes procaspase-1 recruitment and activation, thereby enabling the production and maturation of IL-1β [25]. IL-1β is the result of NLRP3 inflammatory vesicle activation and plays a crucial role in inflammation. Aside from its ability to initiate further inflammatory cascades, IL-1β can also promote macrophage activation by activating IL-1R [26,27]. There is growing evidence that the inflammatory response can be suppressed by reducing IL-1β production through inhibiting NLRP3 activation. We hypothesize that MA could inhibit NLRP3/IL-1β signaling from attenuating the inflammatory response induced by LPS. According to our study, mRNA and protein levels of NLRP3, ASC, and caspase-1 were significantly increased in the LPS-treated group compared to the control group. The NLRP3 signaling pathway was also activated, resulting in notable increases in mRNA and protein levels of IL-18, IL-1β, and significantly increased levels of extracellularly secreted IL-1β, while MA reversed this trend. In addition, the results of NLRP3 gene silencing also suggest that it is a key node in the inflammatory process and that MA can reduce the secretion levels of inflammatory factors such as IL-1β by suppressing NLRP3 levels.

When pro-oxidants and antioxidants are imbalanced, excessive ROS are produced and released. Oxidative stress is closely related to the inflammatory response and numerous studies have shown that ROS are required for activation of the NLRP3 inflammasome and some studies have reported that ROS also influences the initiation phase of the NLRP3 inflammasome [10,28]. Previous studies have shown that LPS induction of inflammation in the NLRP3 inflammasome is accompanied by a significant production of ROS [8,29]. The ROS/ NLRP3/IL-1β signaling axis may be essential in initiating inflammation. Since MA has an antioxidant effect, it is possible that it could inhibit LPS-induced ROS production and thus suppress inflammation through the ROS/NLRP3/IL-1β signaling pathway. In our study, when cells were stimulated with LPS to produce large amounts of ROS, the subsequent activation of NLRP3 led to an increase in IL-1β mRNA and protein expression, raising IL-1β levels. As expected, we found that MA treatment attenuated LPS-induced ROS accumulation in RAW264.7 cells and reduced IL-1β levels. These results suggest that MA may inhibit LPS-induced NLRP3 initiation and activation by reducing ROS levels through attenuating oxidative stress and suppressing inflammatory responses through the ROS/NLRP3/IL-1β signaling axis.

As a major negative regulator of oxidative stress and inflammatory responses, further research on Nrf2 and its related pathways remain important for the clinical management of inflammation-related diseases [30,31]. Activation of the Nrf2 antioxidant pathway has been reported to not only prevent LPS-induced expression of pro-inflammatory cytokines and inflammatory factors but targeting Nrf2 signaling as an effective approach to treating colitis [14,32]. Studies have shown that Nrf2 controls the expression of protective genes, such as HO-1, which has antioxidant and anti-inflammatory properties [33]. As a result of HO-1 expression, the macrophages produce fewer pro-inflammatory cytokines, including TNF-α, IL-6, and IL-1β [32]. As a key target for ROS inhibition, Nrf2 is essential for suppressing ROS-induced inflammatory responses [8]. LPS-induced oxidative stress and inflammation may be due to an unbalanced homeostasis between oxidants and antioxidants. In the present study, we noted a rapid increase in MDA content, a rapid decrease in SOD, GSH-Px, and CAT activity, and a significant increase in ROS content when treated with LPS, indicating that severe oxidative stress and a rapid decrease in antioxidant capacity occur in cells. This trend was inhibited by both MA and NAC. Here, we demonstrate that MA significantly promoted Nrf2 translocation to the nucleus, up-regulated several antioxidant enzyme activities through the Nrf2/HO-1 signaling axis, down-regulated MDA activity, reduced ROS content, inhibited NLRP3 inflammatory factor activation, and alleviated oxidative stress and inflammatory responses in cells. This trend was reinforced in the presence of the Nrf2 activator SFN. Still, when Nrf2 was silenced, the tendency of MA to promote elevated cellular antioxidant enzyme activity and reduce IL-1β secretion was significantly inhibited. These findings suggest that MA may abrogate LPS-induced oxidative stress and inflammatory responses by activating Nrf2 antioxidant signaling.

Notably, MA, as a mixture, contains a variety of antioxidant substances, and there may be multiple pathways that can regulate ROS. For example, many antioxidants are present in the fermentation substrate of MA. Sea buckthorn has a large number of lipophilic antioxidants (mostly tocopherols and carotenoids) as well as hydrophilic antioxidants (flavonoids, phenolic acids, and ascorbic acid) [16]. There are many active compounds in prickly pears, including vitamin C, SOD, and other antioxidants, which means they have a higher antioxidant capacity than other common fruits and vegetables [17]. In addition, the increased amount of homoerotic acid, 1,2,3-trihydroxy benzene, and hydroxyphenyl acetic acid through fermentation has a good antioxidant capacity [18]. Our in vitro results further showed that MA could remove DPPH, ABTS^+^, and OH^-^ from tissues with comparable scavenging capability to V_c_ and that it was able to eliminate superoxide and hydroperoxyl radicals and lipid peroxidation better than MitoQ [18]. According to these findings, MA may be more effective at antioxidant activities than single antioxidants. It has been shown that autophagy in LPS-induced RAW264.7 cells also plays a vital role in ROS scavenging, while MA plays a similar role in ROS scavenging by regulating the autophagy process in CoCl_2_-induced BRL3A cells [5,18]. We can infer that MA may also regulate ROS and thus influence the ROS/NLRP3/IL-1β signaling pathway through the regulation of autophagy.

For the first time, we demonstrate that MA inhibits the ROS/NLRP3/IL-1β signaling axis induced by LPS by enhancing Nrf2-antioxidant signaling in RAW264.7 cells (Figure 6). This study demonstrates that MA can control inflammatory responses at the initiation phase, and therefore, MA may be beneficial for treating inflammation-associated diseases as well as preventing them.

## 4. Materials and Methods

### 4.1. Reagents

Chemicals used in this study were purchased from the following sources: Dulbecco’s modified Eagle’s medium nutrient mix F-12 (DMEM-F121320033), fetal bovine serum (FBS, 10091148),Penicillin streptomycin (15140122) and 0.25% trypsin solution (25095-019) were purchased from GIBCO (Carlsbad, CA, USA); NAC (no. 9165) and LPS (no. L2880) were purchased from Sigma (Saint Louis, MO, USA); 2′,7′-dichlorofluorescein diacetate (DCFH-DA, S0033), cellular CCK-8 kit (C0039), Nitric Oxide Assay Kit (S0021S), TNF-α ELISA kit (PT512), IL-6 ELISA kit (PI326), IL-18 ELISA kit (PI553), IL-1β ELISA kit (PI301), BCA protein concentration assay kit (P0010), reactive oxygen species assay kit (S0033M), cellular nuclear and cytoplasmic protein extraction kit (P0028) were purchased from Beyotime Biotechnology (Shanghai, China). Radiothione (C4733) was purchased from APExBIO (Houston, TX, USA); Superoxide dismutase (SOD, A001-3-2), glutathione peroxidase (GSH-Px, A005-1-2), catalase (CAT, A007-1-1) and malondialdehyde (MDA, A003-1-2) kits were purchased from Nanjing Jiancheng Institute of Biological Engineering (Nanjing, China). RNA extraction kit I was obtained from OMEGA (R6841-01; Norcross, GA, USA); PrimeScrip RT kit was purchased from TaKaRa (RR047A; Ōsaka, Japan); PVDF (No. IPVH00010) was purchased from Millipore(Boston, MA, USA); ECL assay (RPN2232) was purchased from GE Healthcare (Boston, MA, USA). Lipofectamine 2000 (12566014) was purchased from Invitrogen (Carlsbad, CA, USA). MA was purchased from Shanghai Chuangbo Ecological Engineering Co. (Shanghai, China). The specific composition of MA can be found in this reference [18].

### 4.2. Cell Culture and Processing

RAW264.7 cells were purchased from the Cell Bank of the Chinese Academy of Sciences (Shanghai, China) and grown in DMEM-F12 supplemented with 10% (*v/v*) inactivated fetal bovine serum (FBS), 100 U/mL penicillin, and 100 mg/mL streptomycin. Cells were maintained at 37 °C and 5% CO_2_ in a humid environment. MA, NAC, and LPS were all dissolved in the complete medium, then filtered through 0.45 μm Merck Millipore membrane filters and prepared for use.

### 4.3. Cell Viability Assay

Cells (1 × 10^4^ cells/well) were planted in 96-well plate overnight and then treated with different concentrations of MA (0-500 mg/L) for 12 h, followed by stimulation with or without LPS for another 24 h. Inflammation model was constructed using 1 mg/L LPS to stimulate RAW264.7 cells [5]. After discarding supernatant, the treated cells were rinsed three times with DPBS, followed by addition of 100 μL medium and 10 μL CCK-8, and incubated for two hours at 37 °C. An absorbance measurement at 490 nm was conducted, and the results were presented as a fold change compared with the untreated group.

### 4.4. Inflammatory Cytokine Detection

In brief, 1 × 10^4^ RAW264.7 cells/well were pretreated in a 96-well plate with MA (100 mg/L) for 12 h with or without LPS (1 mg/L) for 24 h. The concentrations of inflammatory cytokines including TNF-α, IL-6, IL-18, and IL-1β in the cell supernatant were determined using enzyme-linked immunosorbent assay (ELISA) kits according to the manufacturer’s instructions.

### 4.5. Measurement of Intracellular ROS

The ROS assay kit using DCFH-DA as a fluorescent probe is used to determine the relative levels of intracellular ROS. The cells were processed according to their respective treatments. After removing the supernatant, the cells were washed twice with DPBS. Incubation was conducted for 20 min with 10 μM DCFH-DA probe after treatment. DPBS was then used to wash the cells twice. An excitation/emission wavelength of 535/610 nm was used to read the fluorescence. Fluorescence images were obtained by fluorescence microscopy (Tokyo, Japan).

### 4.6. Determination of Antioxidant Enzyme Activity

The concentration of MDA and the activities of SOD, CAT, and GSH-Px were measured using the respective detection kits (Nanjing Jiancheng Bioengineering Institute, Nanjing, Jiangsu, China) according to the manufacturer’s instructions. The Bradford Protein Assay Kit (Shanghai, China) was used to measure protein concentrations.

### 4.7. Reverese Transcription-Quantitative Real-Time Polymerase Chain Reaction (RT-qPCR)

A total RNA extraction kit was used to extract RNA from cells after treatment. RNA concentration was measured with a spectrophotometer (NanoDrop ND-100, Shanghai Fishing Biotechnology Co., Ltd., Shanghai, China). PrimeScrip RT kit was used to reverse transcribe 1 μg of RNA into cDNA. With the applied Biosystems 7500 real-time PCR system (Thermo, Boston, USA), gene expression was quantified via quantitative real-time PCR reactions. Reactions were carried out in a total volume of 20 μL, including 10 μL SYBR Green mix, 2 μL cDNA, 0.4 μL Rox dye II, 6.8 μL H_2_O and 0.4 μL forward and reverse primers. During the amplification process, the heat was initially applied at 95 °C for 30 s, followed by 40 cycles of 95 °C for 5 s and 60 °C for 34 s, and then gradually increased to 95 °C to obtain melting curve data. Primers were designed based on previous studies. The primer sequences used in this study were shown in Appendix A. As a housekeeping gene, β-actin was used to normalize the transcription of target genes. The relative gene expression was calculated using the 2^−∆∆Ct^ method.

### 4.8. Western Blot Analysis

RAW264.7 cells were inoculated at a density of 1 × 10^5^ cells/mL (10 mL) in Corning cell culture dishes (100 mm × 20 mm, 430167). A BCA Protein Assay Kit was used to measure protein concentrations in whole-cell extracts after cell treatment. A total of 20–40 μg of total protein was electrophoresed with SDS-polyacrylamide gels and then transferred to PVDF membranes. After blocking in Tris-buffered saline/Tween (TBST) containing 5% skimmed milk powder, the membranes were incubated overnight at 4 °C with primary antibodies, including NLRP3 (1:1000, NBP1-31245, Novus, Saint Louis, MO, USA), IL-1β (1:1000, NB600-633, Novus), ASC (1:1000, LS-C40344-100, Lifespan, Saint Louis, MO, USA), Caspase-1 (1:1000, sc-56036, Santacruz, Santa Cruz, CA, USA), IL-18 (1:1000, AF588, R&D Systems, Minneapolis, MN, USA), Nrf2 (1:500, A2019-100, Biovision, San Francisco, CA, USA), Lamin B (1:1000, abs131244, Absin, Shanghai, China), HO-1 (1:1000, abs123962, Absin), NQO1 (1:1000, ab2346, Abcam, Cambridge, UK), GAPDH (1:1000, 5174, CST, Boston, MA, USA). Secondary antibody binding was detected using an anti-rabbit antibody (1:2000, ab97051, Abcam) or anti-mouse IgG HRP (1:2000, sc-2005, Santacruz) and detected by ECL detection reagents. An enhanced chemiluminescence detection system (Tanon, Shanghai, China) was used for image acquisition. Quantification of the density of protein bands was performed using ImageJ V2.6 software.

### 4.9. RNA Silencing

Cells were transfected with negative control NC siRNA, NLRP3 siRNA, or Nrf2 siRNA via customized siRNA reagent systems according to the manufacturer’s instructions. Before carrying out transfection, the cells were grown to 60% confluence. In brief, the cells were seeded in a 6-well culture plate and incubated with the NC siRNA, NLRP3 siRNA, or Nrf2 at 300 nM for 48 h in serum-free OPTI-MEM media (Invitrogen, Carlsbad, CA, USA). The siRNA fragment sequences were shown in Appendix A. After incubation, the transfected cells were pretreated with indicated concentration of MA for 12 h and stimulated with or without LPS (100 ng/mL) for 24 h. The cells were prepared and the expressions of NLRP3 and Nrf2 were analyzed by real-time PCR and Western blot analysis.

### 4.10. Statistical Analysis

All data were analyzed by one-way ANOVA or two-tailed Student’s *t*-test, shown as mean ± SEM. Significance analysis was performed on SPSS 17.0 statistical software (SPSS Inc., Chicago, IL, USA). Values of *p < 0.05* were considered statistically significant.

## Figures and Tables

**Figure 1 ijms-23-12477-f001:**
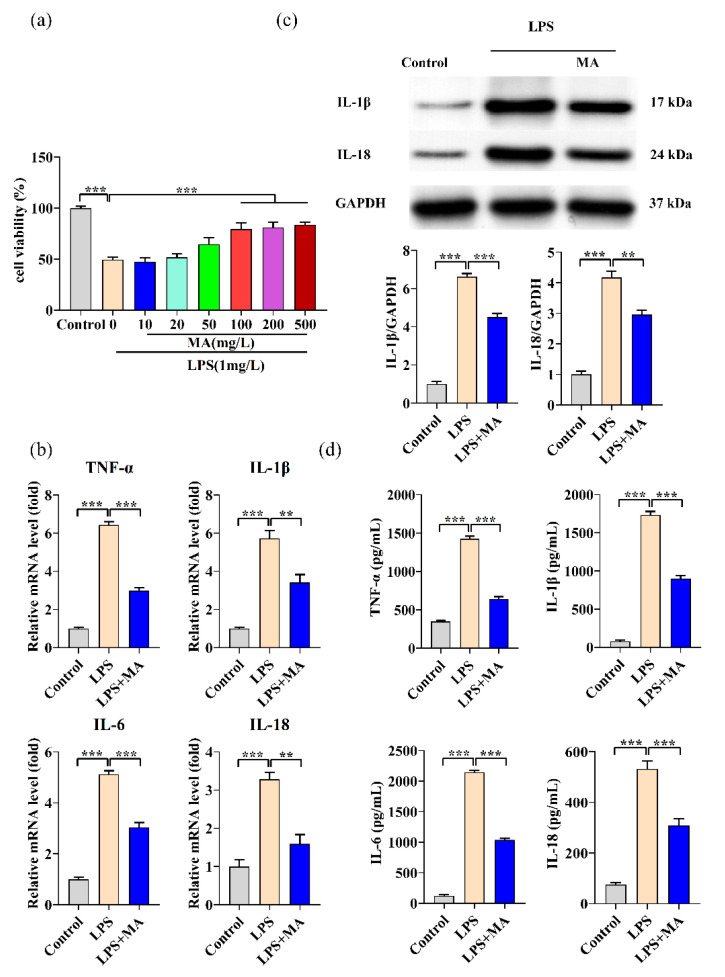
MA exhibits anti-inflammatory effects in LPS-induced RAW264.7 cells. (**a**) Cell viability was assessed by CCK-8 assay in RAW264.7 cells treated with different concentrations of MA (0-500 mg/L) for 12 h, followed by 1 mg/L LPS stimulation for 24 h. (**b**) qRT-PCR was performed to evaluate the mRNA expressions of TNF-α, IL-1β, IL-6, and IL-18 in RAW264.7 cells after treatment with 100 mg/L MA for 12 h, followed by stimulation with 1 mg/L LPS for 24 h. (**c**) Western blot was performed to detect the protein levels of IL-1β and IL-18 in RAW264.7 cells. (**d**) ELISA was conducted to detect the levels of inflammatory cytokines including TNF-α, IL-1β, IL-6, and IL-18 in RAW264.7 cells. Data are shown as mean ± SEM obtained from three independent experiments. ** *p* < 0.01, *** *p* < 0.001.

**Figure 2 ijms-23-12477-f002:**
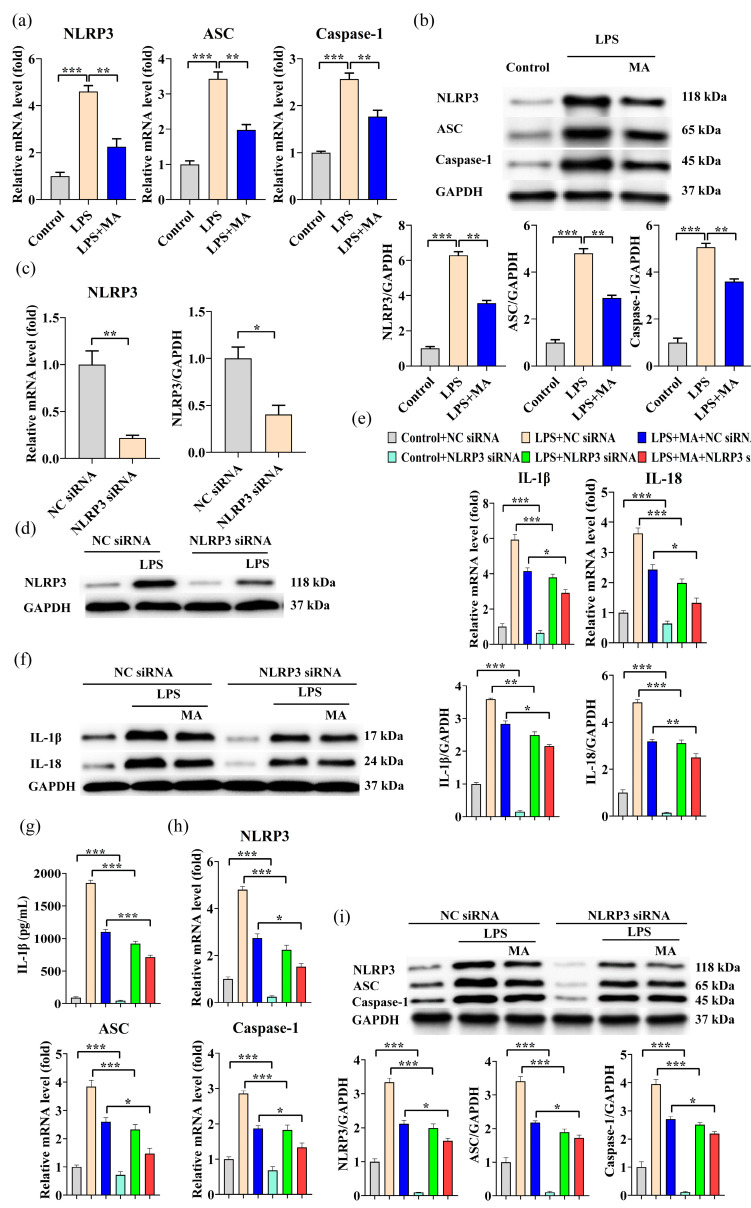
MA attenuates LPS-induced inflammatory effects through inhibition of NLRP3. (**a**) qRT-PCR was performed to evaluate the mRNA expression of NLRP3, ASC and Caspase-1 in RAW264.7 cells. (**b**) Western blot analysis was performed to detect the protein levels of NLRP3, ASC, and Caspase-1 in RAW264.7 cells. (**c**) qRT-PCR was performed to evaluate the mRNA expression of NLRP3 in RAW264.7 cells after siRNA treatment. Cells were first transfected with siRNA for 48 h, followed by treatment with 100mg/L MA for 12h, and finally treated with 1mg/L LPS for 24 h. All subsequent experimental cell treatments in this figure are performed in the same way as this cell treatment. (**d**) Western blot analysis was performed to detect the protein levels of NLRP3 in RAW264.7 cells after siRNA treatment. (**e**) qRT-PCR was performed to evaluate the mRNA expression of IL-1β and IL-18 in RAW264.7 cells after siRNA treatment. (**f**) Western blot was performed to detect the protein levels of IL-1β and IL-18 in RAW264.7 cells after siRNA treatment. (**g**) ELISA was conducted to detect the levels of IL-1β inflammatory cytokines after siRNA treatment. (**h**) qRT-PCR was performed to evaluate the mRNA expression of NLRP3, ASC, and Caspase-1 in RAW264.7 cells after siRNA treatment. (**i**) Western blot was performed to detect the protein levels of NLRP3, ASC, and Caspase-1 in RAW264.7 cells after siRNA treatment. Data are shown as mean ± SEM obtained from three independent experiments. * *p* < 0.05, ** *p* < 0.01, *** *p* < 0.001.

**Figure 3 ijms-23-12477-f003:**
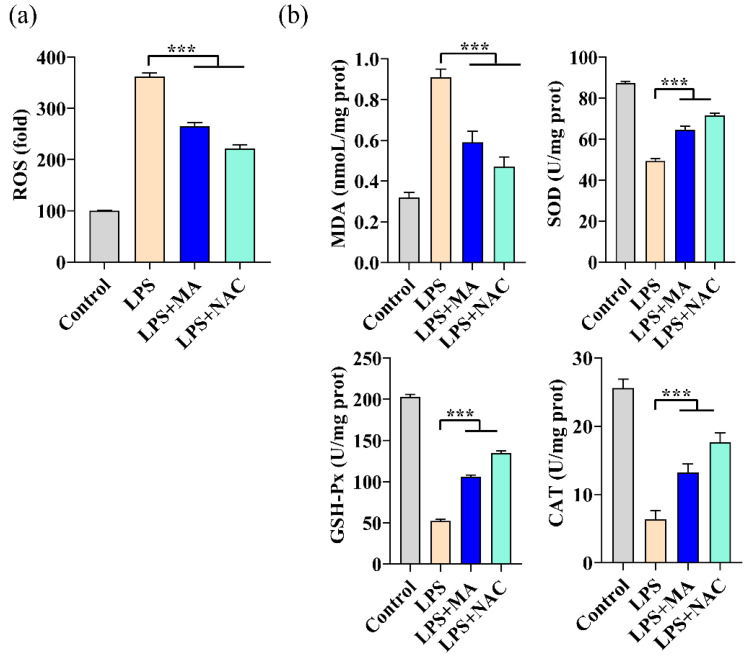
MA mitigates LPS-induced oxidative stress in RAW264.7 cells. Treatment with 100 mg/L MA for 12 h or 10 μM NAC (Positive control) for 30 min, followed by 1 mg/L LPS for 24 h. (**a**) The level of intracellular ROS was measured by DCF fluorescence using an enzyme marker. (**b**) MDA concentration was analyzed with 2-thiobarbituric and the activities of antioxidant enzymes including SOD, GSH-Px, and CAT were determined using ELISA kits. Data are shown as mean ± SEM obtained from three independent experiments. *** *p* < 0.001.

**Figure 4 ijms-23-12477-f004:**
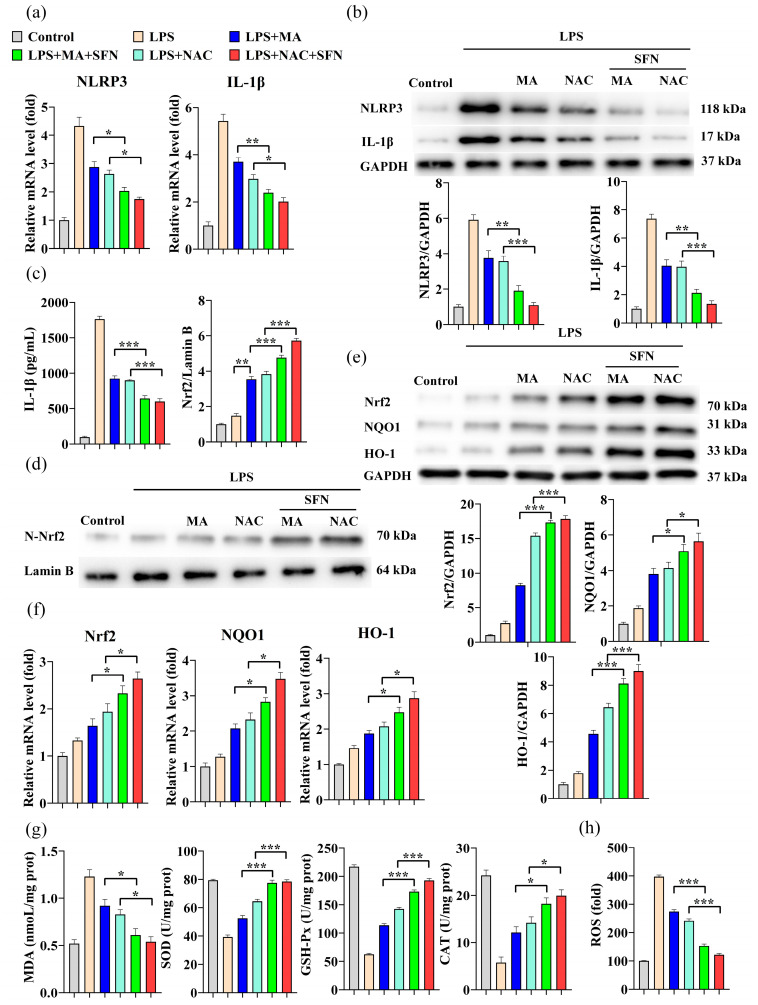
MA attenuates the inflammatory effects induced by LPS through the NLRP3/IL-1β signaling pathway by activating the Nrf2 antioxidant pathway. Cells were pretreated with10 μM SFN for 1 h and/or 100 mg/L MA for 12 h and/or 10 μM NAC (Positive control) for 30 min and then stimulated with LPS (1 mg/L) for 24 h. (**a**) qRT-PCR was performed to evaluate the mRNA expression of NLRP3 and IL-1β in RAW264.7 cells. (**b**) Western blot was performed to detect the protein levels of NLRP3 and IL-1β in RAW264.7 cells. (**c**) ELISA was conducted to detect the levels of inflammatory cytokines IL-1β in RAW264.7 cells. (**d**) Western blot was performed to detect the protein levels of N-Nrf2 (Nuclear-Nrf2) in RAW264.7 cells. (**e**) qRT-PCR was performed to evaluate the mRNA expression of Nrf2, NQO1 and HO-1 in RAW264.7 cells. (**f**) Western blot was performed to detect the protein levels of Nrf2, NQO1, and HO-1 in RAW264.7 cells. (**g**) MDA concentration was analyzed with 2-thiobarbituric and the activities of antioxidant enzymes including SOD, GSH-Px, and CAT were determined using ELISA kits. (**h**) The level of intracellular ROS was measured by DCF fluorescence using an enzyme marker. Data are shown as mean ± SEM obtained from three independent experiments. * *p* < 0.05, ** *p* < 0.01, *** *p* < 0.001.

**Figure 5 ijms-23-12477-f005:**
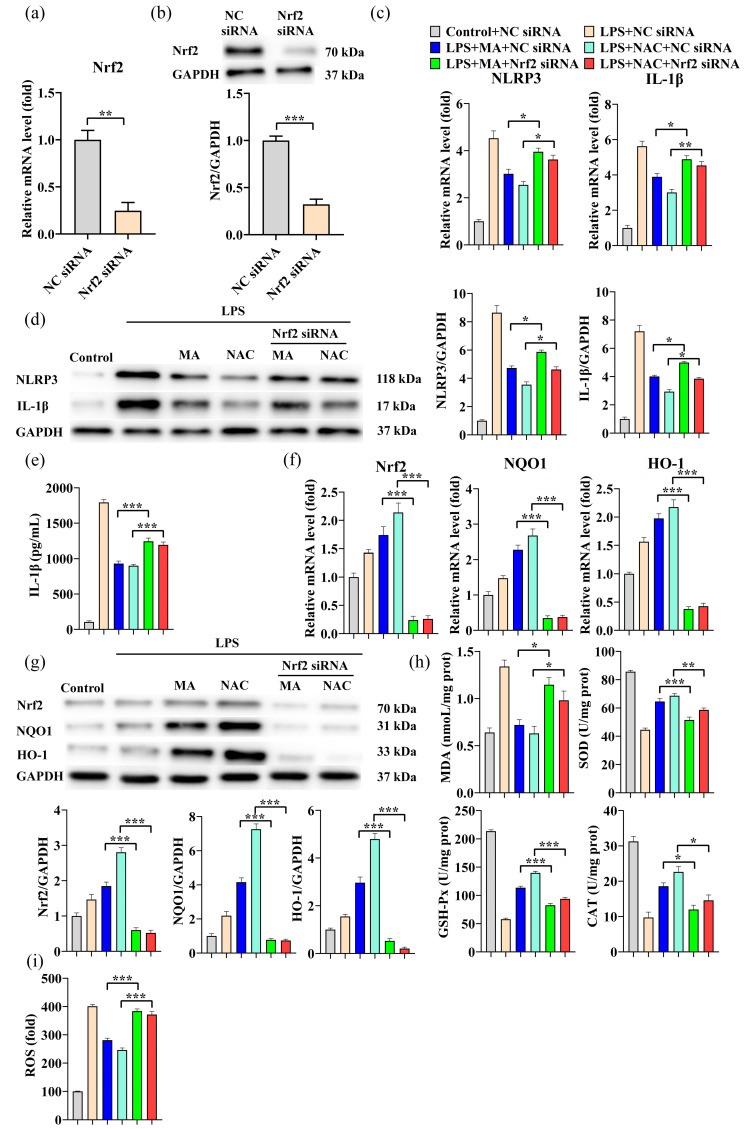
MA attenuates the inflammatory effects induced by LPS through the NLRP3/IL-1β signaling pathway by activating the Nrf2 antioxidant pathway. Cells were first transfected with siRNA for 48 h, followed by treatment with 100 mg/L MA for 12 h or 10μM NAC for 30 min, and finally treated with 1mg/L LPS for 24 h. (**a**) qRT-PCR was performed to evaluate the mRNA expressions of Nrf2 after siRNA treatment. (**b**) Western blot was performed to detect the protein levels of Nrf2 after siRNA treatment. (**c**) qRT-PCR was performed to evaluate the mRNA expressions of NLRP3 and IL-1β after siRNA treatment. (**d**) Western blot was performed to detect the protein levels of NLRP3 and IL-1β after siRNA treatment. (**e**) ELISA was conducted to detect the levels of IL-1β inflammatory cytokines after siRNA treatment. (**f**) qRT-PCR was performed to evaluate the mRNA expressions of Nrf2, HO-1, and NQO1 after siRNA treatment. (**g**) Western blot was performed to detect the protein levels of Nrf2, HO-1, and NQO1 after siRNA treatment. (**h**) MDA concentration was analyzed with 2-thiobarbituric and the activities of antioxidant enzymes including SOD, GSH-Px, and CAT were determined using ELISA kits. (**i**) The level of intracellular ROS was measured by DCF fluorescence using an enzyme marker after siRNA treatment. Data are shown as mean ± SEM obtained from three independent experiments. * *p* < 0.05, ** *p* < 0.01, *** *p* < 0.001.

**Figure 6 ijms-23-12477-f006:**
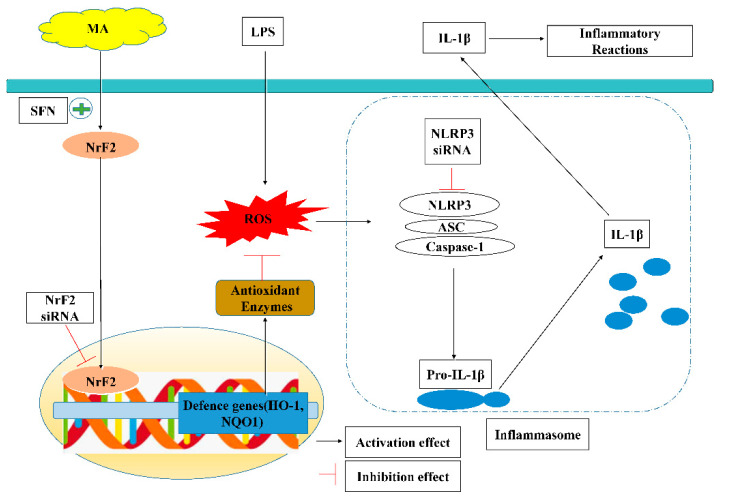
Proposed molecular mechanisms underlying the inhibitory effect of MA on the activation of macrophage induced by LPS. MA activates Nrf2 pathway to increase antioxidant genes and proteins, which in turn reduce inflammation and oxidative stress. The dashed lines are speculations based on previous research.

## Data Availability

Not applicable.

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
