# Peer review of "Microbe-Derived Antioxidants Reduce Lipopolysaccharide-Induced Inflammatory Responses by Activating the Nrf2 Pathway to Inhibit the ROS/NLRP3/IL-1β Signaling Pathway"

_ijms, 2022, doi:10.3390/ijms232012477_

Round 1

Reviewer 1 Report

Inflammation is a major factor for the progression of various chronic diseases/disorders including cardiovascular diseases, cancer, obesity and autoimmune diseases. Natural antioxidant sources are valuable therapeutic agents to reduce illness triggered by oxidative stress. The manuscript “Microbe-induced antioxidants reduce lipopolysaccharide-induced inflammatory response by activating the Nrf2 pathway to inhibit the ROS/NLRP3/IL-1b signaling pathway” by Shen et al. describes the anti-inflammatory effects of the natural antioxidant MA in the context of LPS-induced NLRP3 inflammasome activation. MA reduced ROS level within LPS treated RAW264.7 macrophages influencing NLRP3 inflammasome activation and increased the expression of the antioxidant enzymes SOD, GSH-Px and CAT. siRNA mediated knockdown of Nrf2 abolished the anti-inflammatory and anti-oxidative effect of MA in LPS treated RAW264.7 macrophages.          

The authors use in their study LPS-treated RAW264.7 macrophages to evaluate the effect of MA on NLRP3 inflammasome (consisting of ASC and caspase-1) activation eliciting IL-1b release. It is described by Pelegrin et al. in 2008 (PMID 18490713) and Hirano et al. in 2017 (PMID 28766178) that RAW264.7 macrophages do not express ASC and can only be used to study the transcriptional events regulating the priming process of the NLRP3 inflammasome activation but not the downstream cascade, as they cannot activate the NLRP3 inflammasome complex. ASC is required for the formation of the NLRP3 inflammasome complex finally leading to the secretion of IL-1b. Therefore, the suggested NLRP3 inflammasome activation with the secretion of IL-1b as described by the authors is from a biological point of view not possible. 

RAW264.7 do express TLR4 responsible for the recognition of LPS. Therefore, subsequent activation of RAW264.7 cells might be rather due to the involvement of the NLRC4 or NLRP1b inflammasome. The authors describe the detection of IL-1b in the supernatant of LPS stimulated RAW264.7 macrophages but IL-1b ELISAs do not discriminate between IL-1b and pro-IL-1b. As TLR4 signaling might be active, pro-IL-1b upregulation is functional and is probably released upon cell death of the macrophages. To proof the involvement of NLRP3 inflammasome activation a western blot detecting IL-1b in the supernatant of stimulated RAW264.7 macrophages would have been needed. Therefore, the proposed mechanism of NLRP3 inflammasome involvement in context of the natural antioxidant MA was tested in the wrong model system. An overview of cellular model systems in NLRP3 inflammasome research is published e.g. by Zito et al. in 2020 (PMID 32560261).

Furthermore, figures 2 and 4 are missing in the manuscript. Therefore, an evaluation of the manuscript is not possible. 

Reviewer 2 Report

1. Abstract:

The abstract should be rewritten to increase its readability. There are now some unclear phrases such as "to improve oxidative stress" or "the activation Nrd2 into the nucleus". 

2. Introduction

More extensive information on the properties of MA from previous studies should be included. (Ref 18-21). Now, there is disproportionately little on this subject compared to the general information on inflammation.

3. Materials nad Methods

Subtitle of section 2.7l should be corrected. "Quantitative real-time polymerase chain reaction (qRT-PCR)" should be changed into "reverese transcription-quantitative real-time polymerase chain reaction (RT-qPCR)"

Has the effectiveness of the  qPCR reaction been tested? If the efficiency would be clearly different for the test and reference genes, it should be considered when calculating the relative level of expression.

Were post hoc tests for ANOVA performed? if so which tests were used?

4. Results

Subsection 3.1 - statistical testing of differences in measured parameters (eg. cell viability, protein abundance, gene expression) should be conducted using ANOVA followed by a post hoc test. For example, in Fig 1a, the difference in cell viability should be tested globally considering all concentrations of MA. Then, a post-hoc test should be done between all pairs of concentration of MA. P-values for post hoc tests should be presented.  Now, Fig 1a suggests that concentrations 100, 200, and 500 of MA are considered as one combined group for statistical analysis (horizontal line above the bars 100, 200, and 500).

The same suggestion applies to the other analyzes presented in Fig. 1 and following (ANova plus post-hock test, not Student;s t-test when more than two samples/concentrations are tested). 

It would be helpful to present the bars in the charts in colors rather than in black and white patterns. This would increase readability.

Round 2

Reviewer 1 Report

Priming and activation are two steps required by all inflammasomes to execute its function

Please indicate what priming and activating agents have been used in the study. LPS alone is not sufficient for NLRP3 inflammasome activation. Moreover, as indicated in the material and method section the used LPS in the study is not of ultrapure grade. It is commonly accepted in the inflammasome field, that priming of the NLRP3 inflammasome should be performed with ultrapure LPS, poly(I:C) or Pam3CSK4. Furthermore, the authors should discuss their results under the aspect of the 2 signal paradigm responsible for NLRP3 inflammasome activation (line 606-608).

Using Raw264.7 macrophages to study inflammasome activation is highly debated in the field due to the lackingexpression of ASC in these cell lines. Interestingly, companies use the described lack of ASC in Raw macrophages to verify the specificity of their ASC antibody (e.g. cell signaling)As validation (like STR profiling) in mouse-derived cell lines is difficult I would strongly suggest the authors to validate their key findings in one additional cell line which is more widely accepted to monitor inflammasome activation (like bone marrow derived macrophages, human THP1 cells or human primary monocytes or macrophages).

Furthermore, the manuscript would benefit from showing NLRP3 inflammasome activation by one more additional method than IL-1b release (as it can detect pro-IL-1b as well). The authors can choose from different methods like western blotdetection of IL-1b in the supernatant, ASC specking, western blot to determine caspase-1 cleavage, GSDMD cleavage or usage of reporter cell lines.

Please provide data using the specific NLRP3 inhibitor MCC950or Glybenclamide in addition to the performed siRNA experiments.

How do you explain the influence of the NLRP3 targeting siRNA on the protein level of ASC and caspase-1 (figure 2i)? In the qPCR and ELISA panels no evaluation of control + siNC vs control + siNLRP3 is shown. 

Figure 3a shows different background fluorescence intensities. Please check, that all pictures were taken with the same microscope settings.

Please show in your western blots all bands detected by the antibodies used including the pro- and cleaved form for IL-1b, IL-18 and Caspase-1.

Please indicate the exact LPS within the material and method section. LPS derived from gram-negative bacteria can induce the activation of caspase-11/4/5 inflammasome. Please indicate, if this can be potential mechanism in your study. 

Line 604/609/620: What are NLRP3 inflammatory vesicles? The term is wrong in this context. The cited publication is not referring to NLRP3 induced extracellular vesicles. Please correct.

Please comment on the long priming time with LPS used in the study. More common are priming times for 30min up to 4h.

Do you wash the cells while changing from MA to LPS treatment? Because if you add the LPS to the cells you kind of create a situation of co-incubation of both drugs and you may cause drug-drug interaction in the culture media itself. Under this condition you might not be able to report the effects of any particular drug.

The authors should discuss current NLRP3 inhibitors and compare to MA.
